# Prevalence and Risk Factors of Metabolic Associated Fatty Liver Disease in Xinxiang, China

**DOI:** 10.3390/ijerph17061818

**Published:** 2020-03-11

**Authors:** Hongbin Li, Meihao Guo, Zhen An, Jun Meng, Jing Jiang, Jie Song, Weidong Wu

**Affiliations:** 1School of Public Health, Xinxiang Medical University, Xinxiang 453003, Henan, China; H.health2020@hotmail.com (H.L.); meihao1024@hotmail.com (M.G.); azy1985@163.com (Z.A.); jiangjing2019@sina.com (J.J.); songjie231@126.com (J.S.); 2School of Management, Xinxiang Medical University, Xinxiang 453003, Henan, China; peter.meng1216@hotmail.com

**Keywords:** metabolic associated fatty liver disease, prevalence, risk factors, blood routine test, biochemical examination

## Abstract

Metabolic associated fatty liver disease (MAFLD) is recognized as the liver disease component of metabolic syndrome, which is mainly related to insulin resistance and genetic susceptibility. It is the most prevalent chronic liver disease worldwide. With rapid lifestyle transitions, its prevalence worldwide is increasing, and tremendous challenges in controlling this pandemic are arising. The objective of this study was to investigate the prevalence and risk factors of MAFLD in rural areas of Xinxiang, Henan in 2017. We conducted a cross-sectional analysis of rural inhabitants aged 20–79 years in Xinxiang, Henan in 2017, using cluster random sampling (*N* = 9140). Physical examinations were conducted at local clinics from April to June 2017. After overnight fasting, all participants underwent physical examinations, blood routine tests, biochemical examinations, and liver ultrasound and completed questionnaires. We investigated the crude and age-adjusted MAFLD prevalence and analyzed the characteristics of those with, and without, MAFLD, using logistic regression. Approximately 2868 (31.38%) participants were diagnosed with MAFLD. The overall age-adjusted MAFLD prevalence was 29.85% (men: 35.36%; women: 26.49%). The MAFLD prevalence increased with age, and peaked at the 50–59-year age group, and then began to decline. Higher body mass index, waist circumference, percentage of lymphocytes, levels of hemoglobin, platelet count, triglyceride, fasting plasma glucose, and serum uric acid were independently and positively correlated with MAFLD; In contrary, active physical activity and high-density lipoprotein cholesterol were negatively correlated with MAFLD. In summary, the MAFLD prevalence in the study population was 29.85%. Higher body mass index, waist circumference, percentage of lymphocytes, levels of hemoglobin, platelet count, triglyceride, fasting plasma glucose, and serum uric acid were risk factors for MAFLD.

## 1. Introduction

Metabolic associated fatty liver disease (MAFLD) is recognized as the liver disease component of metabolic syndrome, which is mainly related to insulin resistance and genetic susceptibility [1]. Obesity, Type 2 diabetes mellitus (DM), and metabolic syndrome are consistently identified as the most important risk factors for MAFLD [2].

With rapid lifestyle transitions, the prevalence of MAFLD worldwide is thought to be increasing [3]. Its prevalence in mainland China was 29.81% (27.78–31.93%), while the prevalence in Taiwan was 33.29% (26.42–40.96%) [4]. It is predicted that by 2030, there will be 314.58 million cases of MAFLD in China, which is the fastest growing country in the world [5]; currently, the tremendous challenges in controlling this pandemic are arising.

This was a cross-sectional study including 9140 participants, aiming to determine the prevalence and risk factors of MAFLD in rural areas of Xinxiang, Henan in 2017, through examination of the relationship between routine blood biochemical parameters and the prevalence of MAFLD.

## 2. Materials and Methods

### 2.1. Study Population

This study was conducted in Xinxiang, Henan Province in Central China from April to June 2017 as a part of the baseline investigation of a prospective cohort study on common chronic non-communicable diseases in central China. The location coordinates of Xinxiang city are 113°54′ E 35°18′ N (Figure 1). Stratified random cluster sampling was used in this study. We stratified the economic status of rural areas in Xinxiang and randomly selected two towns, *herein*, Qiliying and Langgongmiao. All permanent residents (only those who had been living in Xinxiang for at least 5 years) at the two sampling sites aged between 20 and 79 years were cluster-sampled, resulting in a sample of 10,280 participants. The following subjects were excluded: (1) Those with mobility problems who cannot attend the physical examination; (2) those who did not answer the questionnaire in a consistent manner; (3) those with incomplete data on physical or biochemical indicators; and (4) those who did not complete liver ultrasound. This led to the exclusion of 1140 individuals from the study, leaving 9140 participants who met our inclusion criteria. The sampling efficiency rate was 88.9%. Among the respondents, 3598 were men, and 5542 women. All subjects gave their informed consent for inclusion before they participated in the study. The study was conducted in accordance with the Declaration of Helsinki, and the protocol was approved by the Ethics Committee of Xinxiang Medical University. The study was conducted in accordance with the Declaration of Helsinki, and the protocol was approved by the Ethics Committee of Xinxiang Medical University for Human Studies (IRB number XY-HS04).

### 2.2. Methods

This study was conducted through physical examinations. A questionnaire focusing on demographic characteristics (e.g., age, sex, and education), behavioral factors, disease history, medication history, and family history, was administered by trained medical workers. Anthropometric measurements, included height, weight, waist circumference, hip circumference, systolic and diastolic blood pressures, and heart rate. Body mass index (BMI) was also calculated. After overnight fasting, the participants underwent routine blood tests, biochemical tests, and liver ultrasound.

Smoking status was categorized as never, former, and current. Current smokers were defined as individuals who reported having smoked at least an average of one cigarette a day during their lifetime and still smoked cigarettes. Former smokers were defined as those who reported having smoked at least an average of one cigarette a day during their lifetime but did not smoke at the time of the interview. Never smokers were defined as those who smoked a maximum of one cigarette a day during their lifetime. Physical activity was assessed via a history of participating in one of the following activities during the past month: (1) Active, performing vigorous physical activity every day (e.g., heavy physical work, fast running, cycling, and climbing); (2) moderately active, performing more than 10 min of moderate exercise every day (e.g., daily housework, childcare, driving, teaching, jogging, dancing, aerobics, and bicycle riding); and (3) inactive, performing no exercise or exercise for no more than 10 min. According to the BMI value, the participants were then assigned to one of three groups: Lean (BMI <23 kg/m^2^), overweight (BMI 23.0–24.9 kg/m^2^), or obese (BMI ≥25.0 kg/m^2^). Excessive alcohol consumption was defined as alcohol consumption of an average of >30 g/day for men and >20 g/day for women [6]. Lean MAFLD is defined as hepatic steatosis with a BMI <23 kg/m^2^ in Asians in the absence of excessive alcohol consumption [7]. Therefore, participants were divided into four groups: Lean MAFLD, overweight MAFLD, obese MAFLD, and non-MAFLD.

Liver ultrasounds were performed by an experienced, well-trained sonographer, using the HITACHI HI VISION Avius ultrasound machine with a 1.0- and 5.0-MHz detector. MAFLD diagnosis was defined according to the Chinese Society of Hepatology. A positive MAFLD diagnosis was made when ultrasound examinations disclosed hepatic steatosis at any stage and excludes other causes of hepatic lipogenesis, such as excessive alcohol consumption [6].

### 2.3. Statistical Analysis

Statistical analysis was performed using Stata software, version 15.1 (StataCorp LP, College Station, TX, United States). We described the prevalence of MAFLD in the study population as a percentage and compared the prevalence differences among different population groups using the chi-square test. As the age distribution of the participants in this study was significantly different from the age distribution of the Chinese population, as reported in the China Population Statistics Yearbook, the 2010 China Population Statistics Yearbook [8] was used for calculating the age-adjusted MAFLD prevalence in the study population.

Behavioral factors, physical examination data and blood parameters of MAFLD group and control group were described. Continuous variables were presented as means ± standard deviations, and Independent sample *t*-test and ANOVA with Bonferroni adjustments for continuous samples were used. Categorical variables were described as proportions, and their differences were compared using the chi-square test.

Collinearity diagnostics of all variables were performed before further statistical analysis. Multivariate logistic regression was used to determine influencing factors for MAFLD. The presence or absence of MAFLD was taken as the dependent variable, and independent variables with significance *p* ≤ 0.05 were introduced in the logistic stepwise regression, excluding the variables with no statistical significance, leaving the factors with independent influence on MAFLD and calculating the odds ratio (OR) and their 95% confidence intervals. *p* values of <0.05 (two-sided) were considered statistically significant. The method used for missing data was a complete case analysis, since statistical packages excluded individuals with any missing value.

## 3. Results

Out of the 9140 participants finally included for the main analyses, 2868 were identified as having MAFLD, 5739 as controls (those without fatty liver disease), and 533 as having viral or drug-induced hepatitis and excessive alcohol consumption. The prevalence of MAFLD in the participants with different characteristics is shown in Table 1. The overall prevalence of MAFLD was 31.38% (29.85% after standardization). The prevalence of MAFLD in the male and female participants was 31.80% and 31.11%, respectively (35.36%, and 26.49%, respectively, after standardization). The overall prevalence of MAFLD increased with age, peaked at the 50–59-year age group, and then began to decline. The prevalence of MAFLD gradually increased with BMI (Table 1).

The participants with MAFLD were more likely to be current smokers, longer duration of sitting, and higher proportion of hypertension, DM, and coronary heart disease and had higher BMI, waist circumference, hip circumference, systolic blood pressure (SBP), and diastolic blood pressure (DBP), but shorter sleep duration than those without fatty liver disease (*p* < 0.05, Table 2). 

The participants with MAFLD also had significantly higher white blood cell (WBC) count, red blood cell (RBC) count, hemoglobin (HGB) level, platelets (PLT) count, alanine aminotransferase (ALT) level, alkaline phosphatase (ALP) level, total bilirubin (TBIL) level, urea level, creatinine (CR) level, triglyceride (TG) level, serum total cholesterol (TC) level, low-density lipoprotein cholesterol (LDL-C) level, fasting plasma glucose (FPG) level, hemoglobin A1c(HBA1c) level, and serum uric acid (SUA) level but lower percentage of monocytes (MONO%), and high-density lipoprotein cholesterol (HDL-C) level (*p* < 0.05, Table 3) than the participants without fatty liver disease. 

A multivariate logistic regression model was employed to identify the influencing factors of MAFLD, and the goodness of fit test showed that the regression model had statistical significance (χ^2^ = 8985.31, *p* < 0.001). The analysis revealed that BMI, waist circumference, LYMPH%, HGB level, PLT level, ALT level, TG level, FPG level, and SUA level were independently and positively correlated with the presence of MAFLD (all *p* < 0.05); in contrary, active physical activity and HDL-C level were independently and negatively correlated with the presence of MAFLD (all *p* < 0.05, Table 4).

Since a large number of participants in this study were non-obese, we compared MAFLD participants with different BMI and the participants without MAFLD (Table 5). The participants in the lean MAFLD group were significantly younger than those in the non-MAFLD group. As BMI of the MAFLD participants increased, the proportion of males decreased, and lean MAFD participants were more likely to be male. The participants with MAFLD were more likely to be current smokers than those without MAFLD, and lean MAFLD group had the most current smokers. Compared with the other two groups of MAFLD participants, the lean MAFLD group had more active activity rates and was similar to the non MAFLD group. With the increase in BMI of the MAFLD participants, the prevalence of hypertension, diabetes and coronary heart disease increased gradually. However, the prevalence of hypertension and coronary heart disease in lean MAFLD participants was lower than that in non-MAFLD participants. In particular, the prevalence of diabetes in lean MAFLD participants was higher than that in non-MAFLD participants. In the comparison of blood biochemical indicators among different groups, except for the difference in RBC level between the lean MAFLD group and the non-MAFLD group, there was no difference in other blood biochemical indicators.

## 4. Discussion

MAFLD is now recognized as the most common liver disease worldwide, affecting nearly one billion people worldwide [9] and its phenotypic manifestations and severity are the result of gene-environment interactions [10]. MAFLD spans a wide spectrum of liver damage severity, ranging from simple steatosis, through non-alcoholic steatohepatitis, to fibrosis and ultimately cirrhosis and hepatocellular carcinoma. Histological changes in most patients are at the stage of simple steatosis [11], with only a few developing chronic inflammation, the harbinger of disease progression, and there is clear evidence of inter-individual variability in all aspects of the natural history [12]. The transition from steatosis to NASH and subsequent liver fibrosis may be multifactorial. Altered microbiota and gut permeability, severity of metabolic alterations, oxidative stress, and pro-inflammatory imbalances may all be involved, with genetic factors playing a major role [13]. The main common genetic risk factor is the PNPLA3 I148M protein variant, which accumulates at the surface of lipid droplets impeding lipid remodeling [14]. However, research on the factors influencing MAFLD, such as genes, were still focused on individuals at risk. Our study was performed at a population level to explore the relationship between routine blood biochemical parameters and the presence of MAFLD, and we hope to identify biomarkers to evaluate the prevalence of MAFLD, identify populations at risk, and provide new strategies for future research.

With the development of urbanization, the number of rural-to-urban migrant workers has been increasing rapidly in China over recent decades [15], and the demographics of rural areas has also changed. Henan is one of the largest provinces in China and has the largest rural population. Henan is a typical labor export province [16], and the economy is less-developed, compared with other provinces. Many young and middle-aged men go out to work, so we have more elderly people in the study population, where women make up a proportion of 60.63%. Our findings indicate that the prevalence of MAFLD in the rural population of Xinxiang, Henan province was 31.38% (29.85% after standardization), which is higher than the average prevalence (20.43%) of MAFLD in rural populations of China [17].

It is well-recognized that increased BMI and waist circumference play a crucial role in the development of MAFLD, and their relationship with MAFLD has been demonstrated in many studies [18,19]. Active exercise was found to reduce intrahepatic lipid contents in patients with MAFLD [20,21]. In addition to weight control, active exercise and adequate sleep, multiple logistic regression analysis showed that the levels of HGB, PLT, LYMPH%, TG, FPG, and SUA were independently and positively correlated with the severity of MAFLD. The liver plays a central role in metabolism of lipids and glucose. Numerous studies have found that patients with MAFLD have increased TC, TG, and LDL-C levels and decreased HDL-C levels [22,23]. Elevated ALT levels were strongly associated with MAFLD and metabolic syndrome [24,25]. High ALT levels were used as a marker of MAFLD [26]. From the perspective of regression coefficient, TG has the greatest influence on MAFLD. In this study, for every 1 μmol/L increase in TG, the probability of MAFLD increased by 0.138 times. A prospective cohort study found that the SUA level was an independent risk factor of incident fatty liver detected by ultrasound [27]. Our study also found an independent positive correlation between the SUA level and MAFLD. The potential pathways behind the association between the SUA level and MAFLD is that SUA might increase the risk of MAFLD directly besides the indirect effects, by increasing the fasting insulin level, blood pressure, and TC level and decreasing the HDL-C level [28]. 

This study found an independent relationship between the HGB level and MAFLD, similar to previous reports that higher HGB levels were independently associated with a higher incidence of MAFLD [29,30,31,32]. Further, the HGB level has been suggested as a simple and reliable biomarker for MAFLD [33]. The progressive form of MAFLD is characterized by inflammation, cellular injury, and fibrosis [34], and peripheral blood leukocyte is often increased. The WBC count is a simple and convenient marker of inflammation. In this study, the WBC count in the MAFLD group was significantly higher than that in the control group. A large-scale health assessment-based longitudinal cohort study in an urban Han Chinese population [35] and a cross-sectional study with Korean adults [36] showed a significant association between the WBC count and the incidence of MAFLD; however, our study did not find an independence effect of the WBC count. In this study, an increased LYMPH% was found to be an independent factor for MAFLD, which may be related to chronic inflammation of the fatty liver. However, we did not find any significant correlation between the presence of MAFLD and other types of white blood cell. Several studies have found the PLT level to be an independent predictor of the severity of fibrosis in MAFLD [37], liver cirrhosis, and non-alcoholic steatohepatitis [38]. Our study found that the subjects with MAFLD had higher PLT levels, which is similar to the findings by Xu et al. [39]. We suggest that this may be attributed to the fact that severe steatosis or advanced fibrosis is relatively rare in the population [11] or that our research indicators were not observed dynamically. Thus, whether PLT values are decreased stepwise when a patient’s condition progressively deteriorates remains unclear, and further prospective studies are needed.

MAFLD is strongly associated with obesity, however, it can also occur in non-obese people, initially described as “lean MAFLD” in Asian populations. It was recognized that between 5% and 45% of patients with MAFLD were lean [7], and in this study, 6.24% of MAFLD participants were lean. Our results suggested that, compared with participants who have MAFLD and were overweight or obese, lean MAFLD participants were younger, more likely to be male, and have a lower prevalence of the metabolic syndrome, which is similar to previous reports [40]. The active activity rate of the lean MAFLD group in this study was similar to the non MAFLD group, and was higher than that of the other two groups of MAFLD participants. The participants in the lean MAFLD group had a lower BMI than those in the non-MAFLD group. This indicates that, although the BMI of the lean MAFLD group was lower than that of the non-MAFLD group, and participants in the lean MAFLD group were as active as those in the non-MAFLD group, the prevalence of diabetes in the lean MAFLD group was higher than that in the non-MAFLD group. This is similar to previous reports that patients with MAFLD who are lean are usually insulin resistant when compared with matched controls without MAFLD [41]. In addition, there were more current smokers in the lean MAFLD group, and smoking may be a risk factor for MAFLD in lean people. RBC levels in the lean MAFLD group were higher than those in the non-MAFLD group, but were similar to participants who have MAFLD and were overweight or obese. Elevated RBC levels may also be associated with the presence of MAFLD in lean persons. 

In this study, the prevalence of diabetes in the lean MAFLD group was higher than that in the non-MAFLD group. We speculate that this may be related to genes, and there was evidence for shared genetic modifiers that link MAFLD and related metabolic disorders [10]. Our study cannot prove why the lean MAFLD group had a higher prevalence of diabetes, only to find that the lean MAFLD group had a higher proportion of current smokers and higher RBC levels. Previous studies have shown that fecal and blood microbiota profiles showed different patterns between subjects with obese and lean MAFLD [42]. Increased visceral obesity (as opposed to general obesity), characteristic fat intake [43], and genetic risk factors [44], including congenital defects of metabolism, might be associated with lean MAFLD [9]. Further studies are needed to prove mechanisms of lean MAFLD.

The strength of this study is that it was conducted in a large rural population. Further, only a few studies have reported the blood biochemical parameters of MAFLD in the general population. The study findings can be used as a corroboration of the blood biochemical results of a small population.

This study also has several limitations. First, the cross-sectional design cannot identify any causal relationship between the influencing factors and MAFLD. Second, living habits, such as drinking habits, were assessed on the basis of the participants’ self-assessments, which may result in recall bias. Finally, MAFLD was diagnosed using ultrasound methods instead of histological assessments; nevertheless, ultrasound methods are widely used for population-based studies.

## 5. Conclusions

In conclusion, our findings showed that the prevalence of MAFLD in the study population was 31.38% (29.85% after standardization). Higher body mass index, waist circumference, percentage of lymphocytes, levels of hemoglobin, platelet count, triglyceride, fasting plasma glucose, and serum uric acid were risk factors for MAFLD.

## Figures and Tables

**Figure 1 ijerph-17-01818-f001:**
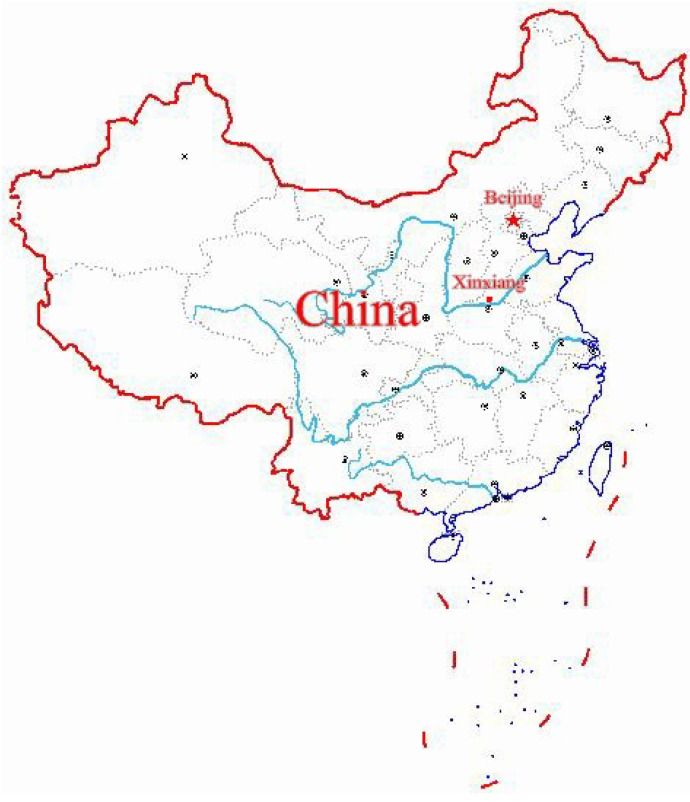
The location of Xinxiang city in China.

**Table 1 ijerph-17-01818-t001:** MAFLD prevalence stratified by different subgroups.

Subgroups	Total Number of Participants	Total Number of Participants with MAFLD	MAFLD Prevalence	*χ^2^*	*p* Value
Overall	9140	2868	31.38%		
Sex					
Male	3598	1144	31.80%	0.4789	0.489
Female	5542	1724	31.11%		
Age at recruitment, years					
20–29	540	138	25.56%	32.2759	<0.001
30–39	955	267	27.96%		
40–49	1909	601	31.48%		
50–59	2299	790	34.36%		
60–69	2523	823	32.62%		
70–79	914	249	27.24%		
Body mass index, kg/m^2^					
<23.0	2288	179	7.82%	1.3e+03	<0.001
23.0–24.9	1888	393	20.82%		
≥25.0	4964	2296	46.25%		

MAFLD: metabolic associated fatty liver disease.

**Table 2 ijerph-17-01818-t002:** Comparison of the behavioral factors and physical examination data between the participants with and without MAFLD.

Behavioral and Clinical Status	MAFLD (*n* = 2868)	Control (*n* = 5739)	*t/χ* ^2^	*p* Value
Smoking status				
Never	2146 (32.84%)	4388 (67.16%)	8.9727	0.011
Former	183 (36.49%)	410 (63.51%)		
Current	539 (30.86%)	938 (69.14%)		
Physical activity				
Inactive	344 (38.48%)	500 (61.52%)	73.9346	<0.001
Moderately active	1841 (35.70%)	3316 (64.30%)		
Active	683 (26.72%)	1873 (73.28%)		
Hypertension				
Yes	905 (40.66%)	1321 (59.34%)	73.5523	<0.001
No	1946 (30.70%)	4393 (69.30%)		
Diabetes mellitus				
Yes	282 (50.36%)	278 (49.64%)	78.7159	<0.001
No	2564 (32.08%)	5428 (67.92%)		
Coronary heart disease				
Yes	303 (40.03%)	454 (59.97%)	17.0507	<0.001
No	2543 (32.62%)	5253 (67.38%)		
BMI, kg/m^2^	27.79 ± 3.48	24.30 ± 3.17	46.4583	<0.001
Waist circumference in men, cm	95.03 ± 8.79	85.55 ± 9.02	28.5894	<0.001
Waist circumference in women, cm	92.10 ± 9.78	81.79 ± 9.37	37.2013	<0.001
Hip circumference, cm	100.33 ± 6.88	95.50 ± 6.34	32.3164	<0.001
SBP, mmHg	134.93 ± 19.48	129.40 ± 20.56	11.9541	<0.001
DBP, mmHg	84.11 ± 11.03	80.25 ± 11.39	14.9791	<0.001

Data were expressed as n (%) or means (standard deviations), as appropriate. MAFLD: metabolic associated fatty liver disease; BMI: body mass index; SBP: systolic blood pressure; DBP: diastolic blood pressure.

**Table 3 ijerph-17-01818-t003:** Comparison of the blood biochemical parameters between the participants with and without MAFLD.

Blood Parameters	MAFLD	Control	*t*	*p* Value
WBC, ×10^9^/L	6.30 ± 1.63	5.82 ± 1.57	13.0958	<0.001
RBC, ×10^12^/L	4.86 ± 0.47	4.72 ± 0.46	13.1568	<0.001
HGB, g/L	142.54 ± 15.93	138.17 ± 16.72	11.5679	<0.001
PLT, ×10^9^/L	247.11 ± 63.43	238.42 ± 63.68	5.9598	<0.001
NEUT%, %	58.26 ± 8.09	58.49 ± 8.59	−1.1907	0.234
LYMPH%, %	33.88 ± 7.59	33.56 ± 7.99	1.8054	0.071
MONO%, %	5.43 ± 1.32	5.56 ± 1.44	−4.2614	<0.001
EO%, %	2.01 ± 1.70	1.97 ± 1.93	1.0541	0.292
BASO%, %	0.42 ± 0.30	0.42 ± 0.33	−0.0335	0.973
ALT, U/L	24.90 ± 17.89	20.38 ± 20.96	9.8647	<0.001
AST, U/L	23.38 ± 8.55	22.97 ± 11.85	1.6300	0.103
ALP, U/L	87.37 ± 24.71	86.01 ± 26.01	2.3148	0.021
TBIL, μmol/L	16.64 ± 7.33	16.35 ± 7.58	1.7072	0.088
Urea, mmol/L	5.10 ± 1.38	5.03 ± 1.45	2.3892	0.017
CR, μmol/L	62.73 ± 13.60	61.71 ± 14.13	3.1972	0.001
TG, mmol/L	2.10 ± 1.58	1.46 ± 1.08	22.3593	<0.001
TC, mmol/L	5.38 ± 1.05	5.17 ± 1.03	8.9123	<0.001
HDL-C, mmol/L	1.17 ± 0.27	1.33 ± 0.31	−23.7332	<0.001
LDL-C, mmol/L	3.09 ± 0.85	2.93 ± 0.85	8.5616	<0.001
SUA, μmol/L	314.05 ± 86.62	277.21 ± 76.86	20.0556	<0.001
FPG, mmol/L	6.12 ± 1.82	5.63 ± 1.42	13.5223	<0.001
HBA1c, %	5.96 ± 1.10	5.64 ± 0.89	14.6705	<0.001

MAFLD: Metabolic associated fatty liver disease; WBC: white blood cell; RBC: red blood cell; HGB: hemoglobin; PLT: platelets; NEUT%: percentage of neutrophils; LYMPH%: percentage of lymphocytes; MONO%: percentage of monocytes; EO%: percentage of eosinophils; BASO%: percentage of basophils; ALT: alanine aminotransferase; AST: aspartate aminotransferase; ALP: alkaline phosphatase; TBIL: total bilirubin; CR: creatinine; TG: triglyceride; TC: serum total cholesterol; HDL-C: high-density lipoprotein cholesterol; LDL-C: low-density lipoprotein cholesterol; SUA: serum uric acid; FPG: fasting plasma glucose; HBA1c: hemoglobin A1c.

**Table 4 ijerph-17-01818-t004:** Multivariate analysis of risk factors for MAFLD.

Variables	Coefficient	Standard Error	Odds Ratio (95%CI)	*p* Value
Physical Activity				
Moderately active	−0.011	0.090	0.988 (0.829–1.179)	0.898
Active	−0.446	0.097	0.640 (0.529–0.775)	<0.001
BMI, kg/m^2^	0.175	0.014	1.192 (1.160–1.224)	<0.001
Waist circumference, cm	0.049	0.005	1.050 (1.040–1.060)	<0.001
HGB, g/L	0.007	0.002	1.007 (1.004–1.011)	<0.001
PLT, ×10^9^/L	0.002	<0.001	1.002 (1.002–1.003)	<0.001
LYMPH%,%	0.012	0.004	1.012 (1.005–1.019)	<0.001
TG, μmol/L	0.138	0.025	1.148 (1.092–1.207)	<0.001
HDL-C, mmol/L	−0.648	0.106	0.523 (0.425–0.644)	<0.001
FPG, mmol/L	0.104	0.017	1.110 (1.073–1.148)	<0.001
SUA, μmol/L	0.002	<0.001	1.002 (1.001–1.003)	<0.001

MAFLD: metabolic associated fatty liver disease; BMI: body mass index; HGB: hemoglobin; PLT: platelet count; LYMPH%: percentage of lymphocytes; TG: triglyceride; HDL-C: high-density lipoprotein cholesterol; FPG: fasting plasma glucose; SUA: serum uric acid.

**Table 5 ijerph-17-01818-t005:** Clinical characteristics of participants with MAFLD groups versus control group.

Characteristics	Lean MAFLD (*n* = 179)	Overweight MAFLD (*n* = 393)	Obese MAFLD (*n* = 2296)	Non-MAFLD (*n* = 5739)	*F/χ* ^2^	*p* Value
Age	50.39 ± 13.88 *	53.89 ± 11.91	53.92 ± 12.47	53.26 ± 13.58	4.6998	0.003
Sex						
Male	79 (44.13%)	158 (40.20%)	907 (39.50%)	2012 (35.06%)	20.7641	<0.001
Female	100 (55.87%)	235 (59.80%)	1389 (60.50%)	3727 (64.94%)		
Smoking status						
Never	124 (69.27%)	296 (75.32%)	1726 (75.17%)	4388 (76.5%)	15.9921	0.014
Former	13 (7.26%)	17 (4.33%)	153 (6.66%)	410 (7.15%)		
Current	42 (23.46%)	80 (20.36%)	417 (18.16%)	938 (16.35%)		
Physical activity						
Inactive	21 (11.73%)	46 (11.70%)	277 (12.06%)	550 (9.58%)	77.7723	<0.001
Moderately active	104 (58.1%)	255 (64.89%)	1482 (64.55%)	3316 (57.78%)		
Active	54 (30.17%)	92 (23.41%)	537 (23.39%)	1873 (32.64%)		
Hypertension						
Yes	32 (17.88%)	84 (21.54%)	789 (34.57%)	1321 (23.12%)	122.0723	<0.001
No	147 (82.12%)	306 (78.46%)	1493 (65.43%)	4393 (76.88%)		
Diabetes mellitus						
Yes	16 (8.99%)	30 (7.71%)	236 (10.36%)	278 (4.87%)	82.7725	<0.001
No	162 (91.01%)	359 (92.29%)	2043 (89.64%)	5428 (95.13%)		
Coronary heart disease						
Yes	8 (4.49%)	34 (8.74%)	261 (11.45%)	454 (7.96%)	28.9884	<0.001
No	170 (95.51%)	355 (91.26%)	2018 (88.55%)	5253 (92.04%)		
BMI, kg/m^2^	21.68 ± 1.11 *	24.16 ± 0.55 *	28.89 ± 2.92 *	24.30 ± 3.17	1396.4295	<0.001
Waist circumference in men, cm	80.16 ± 5.94 *	87.20 ± 4.08	97.69 ± 7.37 *	85.55 ± 9.02	476.0183	<0.001
Waist circumference in women, cm	76.18 ± 5.35 *	83.81 ± 4.98 *	94.66 ± 8.72 *	81.79 ± 9.37	722.0930	<0.001
Hip circumference, cm	92.11 ± 6.22 *	94.94 ± 4.37	101.90 ± 6.34 *	95.50 ± 6.34	626.0271	<0.001
SBP, mmHg	126.61 ± 20.55	131.26 ± 18.83	136.21 ± 19.27 *	129.40 ± 20.56	65.4556	<0.001
DBP, mmHg	78.37 ± 10.93	81.06 ± 10.42	85.08 ± 10.91 *	80.25 ± 11.39	106.5924	<0.001
WBC, ×10^9^/L	6.00 ± 1.54	6.04 ± 1.65 *	6.36 ± 1.62 *	5.82 ± 1.57	64.0206	<0.001
RBC, ×10^12^/L	4.82 ± 0.46 *	4.82 ± 0.48 *	4.87 ± 0.47 *	4.72 ± 0.46	59.5651	<0.001
HGB, g/L	141.06 ± 17.56	142.12 ± 16.30 *	142.73 ± 15.74 *	138.17 ± 16.72	45.2652	<0.001
PLT, ×10^9^/L	236.16 ± 53.67	245.17 ± 61.79	248.29 ± 64.34 *	238.42 ± 63.68	13.9895	<0.001
NEUT%, %	58.36 ± 8.49	58.41 ± 8.84	58.23 ± 7.93	58.49 ± 8.59	0.5334	0.659
LYMPH%, %	33.52 ± 7.88	33.87 ± 8.20	33.91 ± 7.47	33.56 ± 7.99	1.2242	0.299
MONO%, %	5.72 ± 1.57	5.31 ± 1.24 *	5.42 ± 1.31 *	5.56 ± 1.44	9.4631	<0.001
EO%, %	2.00 ± 2.09	1.98 ± 1.65	2.02 ± 1.67	1.97 ± 1.93	0.4096	0.746
BASO%, %	0.39 ± 0.24	0.43 ± 0.29	0.42 ± 0.30	0.42 ± 0.33	0.5844	0.625
ALT, U/L	19.29 ± 10.40	22.19 ± 12.37	25.80 ± 18.99 *	20.38 ± 20.96	41.2049	<0.001
AST, U/L	21.89 ± 6.13	23.02 ± 6.65	23.55 ± 8.98	22.97 ± 11.85	2.3488	0.071
ALP, U/L	83.70 ± 23.85	88.40 ± 25.42	87.48 ± 24.64	86.01 ± 26.01	3.2391	0.021
TBIL, μmol/L	16.82 ± 8.06	17.26 ± 7.55	16.52 ± 7.23	16.35 ± 7.58	2.0928	0.099
Urea, mmol/L	5.17 ± 1.48	5.08 ± 1.41	5.10 ± 1.37	5.03 ± 1.45	2.0733	0.102
CR, μmol/L	61.68 ± 13.09	61.88 ± 12.90	62.96 ± 13.75 *	61.71 ± 14.13	4.4289	0.004
TG, mmol/L	1.61 ± 1.41	1.98 ± 1.35 *	2.16 ± 1.62 *	1.46 ± 1.08	179.3021	<0.001
TC, mmol/L	5.14 ± 0.99	5.35 ± 1.04 *	5.41 ± 1.06 *	5.17 ± 1.03	30.1954	<0.001
HDL-C, mmol/L	1.32 ± 0.33	1.20 ± 0.28 *	1.15 ± 0.26 *	1.33 ± 0.31	208.2637	<0.001
LDL-C, mmol/L	2.83 ± 0.82	3.00 ± 0.81	3.13 ± 0.86 *	2.93 ± 0.85	32.7975	<0.001
SUA, μmol/L	287.59 ± 102.28	290.05 ± 77.64 *	320.21 ± 85.68 *	277.21 ± 76.86	157.9704	<0.001
FPG, mmol/L	5.89 ± 2.09	5.92 ± 1.75 *	6.17 ± 1.81 *	5.63 ± 1.42	65.1338	<0.001
HBA1c, %	5.83 ± 1.10	5.85 ± 1.06 *	5.99 ± 1.10 *	5.64 ± 0.89	75.5209	<0.001

Data were expressed as n (%) or means (standard deviations), as appropriate. * Compared with non-MAFLD group by ANOVA with Bonferroni adjustments, the difference was significant; MAFLD: metabolic associated fatty liver disease; BMI: body mass index; SBP: systolic blood pressure; DBP: diastolic blood pressure; WBC: white blood cell; RBC: red blood cell; HGB: hemoglobin; PLT: platelets; NEUT%: percentage of neutrophils; LYMPH%: percentage of lymphocytes; MONO%: percentage of monocytes; EO%: percentage of eosinophils; BASO%: percentage of basophils; ALT: alanine aminotransferase; AST: aspartate aminotransferase; ALP: alkaline phosphatase; TBIL: total bilirubin; CR: creatinine; TG: triglyceride; TC: serum total cholesterol; HDL-C: high-density lipoprotein cholesterol; LDL-C: low-density lipoprotein cholesterol; SUA: serum uric acid; FPG: fasting plasma glucose; HBA1c: hemoglobin A1c.

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
