# Peer review of "Prevalence and Risk Factors of Metabolic Associated Fatty Liver Disease in Xinxiang, China"

_ijerph, 2020, doi:10.3390/ijerph17061818_

Round 1

Reviewer 1 Report

It's a good idea investigate the prevalence and risk factors of NAFLD in rural areas of Xinxiang, Henan, in which authors had found that the positive association of NAFLD prevalence with age, body mass index, waist circumference, triglyceride, serum glucose and uric acid. However, regarding to the new strategies to prevent NAFLD, the underlying mechanism still not very clear. I would like to ask few questions

  1. What's different between patients from rural areas of Xinxiang, Henan and other rural areas of china. I would be nice to present something are relative new and identic for the local area 
  2. It'd be very nice to have a better or identic biomarker to evaluate the prevalence and future new strategies.
  3. more details for statistical analysis, need to be improved

Author Response

Point 1: What's different between patients from rural areas of Xinxiang, Henan and other rural areas of china. I would be nice to present something are relative new and identic for the local area.

Response 1: Thanks for the Reviewer’s comment. The difference between the patients from rural areas of Xinxiang, Henan and other rural areas of china have been described in the Discussion section of revised manuscript on Page 11, lines 236-246.

Point 2: It'd be very nice to have a better or identic biomarker to evaluate the prevalence and future new strategies.

Response 2: We accept the reviewer’s suggestion and will try to use specific and novel biomarkers and new strategies to evaluate the prevalence of diseases.

Point 3: more details for statistical analysis, need to be improved.

Response 3: Following the Reviewer’s suggestion, detailed statistical analyses are added to the Statistical analysis section on Page 4, lines 112-132.

Reviewer 2 Report

The study by Hongbin Li is interesting and provide an an important information on the prevalence of fatty liver disease in one of largest Chinese city.  I am very positive on this work and have some comments that will help strengthen the paper. Please find my comments:

1.       There is a considerable proportion of the patients are non-obese. However, there is a lack of any discussion on this. Authors should discuss the prevalence, mechanisms of lean MAFLD in the context of literature and cite for example these references (Nat Rev Gastroenterol Hepatol. 2018 Jan;15(1):11-20, Lancet Gastroenterol Hepatol. 2020 Feb;5(2):167-228,  Scand J Gastroenterol. 2009;44(4):471-7, Nutr Metab Cardiovasc Dis. 2018 Apr;28(4):369-384, Hepatology. 2019 Aug 23. doi: 10.1002/hep.30908, PLoS One. 2019 Mar 14;14(3):e0213692, Aliment Pharmacol Ther. 2018 Dec;48(11-12):1260-1270).

2.       It would be interesting to add another table, similar to table 2, but for the clinical characteristics according to lean MAFLD.

3.       Figure 1 is redundant as the same data are presented in the table, I would suggest delete this figure.

4.       There is a lack of discussion of role of genetics in MAFLD, which is very important to discuss briefly. Here is some examples of relevant key reviews that can help they authors to do this and they can also cite it (J Hepatol. 2016 Aug;65(2):399-412, J Hepatol. 2018 Feb;68(2):268-279, Curr Pharm Des. 2018;24(38):4566-4573. Nat Rev Gastroenterol Hepatol. 2020 Jan;17(1):40-52).

5.       According to a recent international consensus including a key opinion Chinese leaders the nomenclature of NAFLD has been updated to metabolic associated fatty liver disease (MAFLD). Please, update the manuscript accordingly and refer to the reference. (MAFLD: A consensus-driven proposed nomenclature for metabolic associated fatty liver disease. Gastroenterology. 2020 Feb 7. pii: S0016-5085(20)30171-2. doi: 10.1053/j.gastro.2019.11.312).

6.       Adding map to China showing the location of Xinxiang, Henan, may be useful in the visual presentation.

Author Response

Point 1: There is a considerable proportion of the patients are non-obese. However, there is a lack of any discussion on this. Authors should discuss the prevalence, mechanisms of lean MAFLD in the context of literature and cite for example these references (Nat Rev Gastroenterol Hepatol. 2018 Jan;15(1):11-20, Lancet Gastroenterol Hepatol. 2020 Feb;5(2):167-228,  Scand J Gastroenterol. 2009;44(4):471-7, Nutr Metab Cardiovasc Dis. 2018 Apr;28(4):369-384, Hepatology. 2019 Aug 23. doi: 10.1002/hep.30908, PLoS One. 2019 Mar 14;14(3):e0213692, Aliment Pharmacol Ther. 2018 Dec;48(11-12):1260-1270).

Response 1: We greatly appreciate the Reviewer’s suggestion and help. In the revised manuscript, we have discussed the prevalence, mechanisms of lean MAFLD on Page 12, lines 303-330.  

Point 2: It would be interesting to add another table, similar to table 2, but for the clinical characteristics according to lean MAFLD.

Response2: According to the Reviewer’s suggestion, we add the new Table 5 in the revised manuscript on Page 8-10.

Point 3: Figure 1 is redundant as the same data are presented in the table, I would suggest delete this figure.

Response 3: Thanks for the Reviewer’s insightful comment, Figure 1 in the old version of manuscript is deleted in the revised manuscript. 

Point 4: There is a lack of discussion of role of genetics in MAFLD, which is very important to discuss briefly. Here is some examples of relevant key reviews that can help they authors to do this and they can also cite it (J Hepatol. 2016 Aug;65(2):399-412, J Hepatol. 2018 Feb;68(2):268-279, Curr Pharm Des. 2018;24(38):4566-4573. Nat Rev Gastroenterol Hepatol. 2020 Jan;17(1):40-52).

Response 4: We greatly appreciate the Reviewer’s help in improving this manuscript. According to the Reviewer’s suggestion, we discuss the role of genetics in MAFLD in the Discussion section on Page 10-11, lines227-235. 

Point 5: According to a recent international consensus including a key opinion Chinese leaders the nomenclature of NAFLD has been updated to metabolic associated fatty liver disease (MAFLD). Please, update the manuscript accordingly and refer to the reference. (MAFLD: A consensus-driven proposed nomenclature for metabolic associated fatty liver disease. Gastroenterology. 2020 Feb 7. pii: S0016-5085(20)30171-2. doi: 10.1053/j.gastro.2019.11.312).

Response 5: We again thank the Reviewer for the updates. NAFLD in the old version has been replaced with MAFLD, where appropriate.

Point 6: Adding map to China showing the location of Xinxiang, Henan, may be useful in the visual presentation.

Response 6: According to the reviewer’s suggestion, the location of Xinxiang has been labeled on China MAP as Figure in the revised manuscript on Page 3.

Round 2

Reviewer 1 Report

accept current version